# Differences in Anthropometric and Clinical Features among Preschoolers, School-Age Children, and Adolescents with Obstructive Sleep Apnea—A Hospital-Based Study in Taiwan

**DOI:** 10.3390/ijerph17134663

**Published:** 2020-06-29

**Authors:** Hai-Hua Chuang, Jen-Fu Hsu, Li-Pang Chuang, Ning-Hung Chen, Yu-Shu Huang, Hsueh-Yu Li, Jau-Yuan Chen, Li-Ang Lee, Chung-Guei Huang

**Affiliations:** 1Department of Family Medicine, Chang Gung Memorial Hospital, Taipei and Linkou Branches, Taoyuan 33305, Taiwan; chhaihua@gmail.com (H.-H.C.); welins@cgmh.org.tw (J.-Y.C.); 2Department of Industrial Engineering and Management, National Taipei University of Technology, Taipei 10608, Taiwan; 3Obesity Institute, Genomic Medicine Institute, Geisinger, Danville, 17822 PA, USA; 4College of Medicine, Chang Gung University, Taoyuan 33302, Taiwan; jeff0724@gmail.com (J.-F.H.); r5243@cgmh.org.tw (L.-P.C.); ninghung@cgmh.org.tw (N.-H.C.); yushuhuang1212@gmail.com (Y.-S.H.); hyli38@cgmh.org.tw (H.-Y.L.); 5Department of Pediatrics, Chang Gung Memorial Hospital, Linkou Branch, 33305 Taoyuan, Taiwan; 6Department of Pulmonary and Critical Care Medicine, Sleep Center, Chang Gung Memorial Hospital, Linkou Branch, Taoyuan 33305, Taiwan; 7Department of Child Psychiatry, Sleep Center, Chang Gung Memorial Hospital, Linkou Branch, Taoyuan 33305, Taiwan; 8Department of Otorhinolaryngology-Head and Neck Surgery, Sleep Center, Chang Gung Memorial Hospital, Linkou Branch, Taoyuan 33305, Taiwan; 9Department of Laboratory Medicine, Chang Gung Memorial Hospital, Linkou Branch, Taoyuan 33305, Taiwan; 10Department of Medical Biotechnology and Laboratory Science, Graduate Institute of Biomedical Sciences, Chang Gung University, Taoyuan 33302, Taiwan

**Keywords:** adolescents, anthropometrics, children, disease severity, gender difference, inflammation, obstructive sleep apnea

## Abstract

Pediatric obstructive sleep apnea (OSA) is associated with adverse health outcomes; however, little is known about the diversity of this population. This retrospective study aims to investigate age-related differences in the anthropometric and clinical features of this population. A total of 253 Taiwanese children (70 (27.7%) girls and 183 (72.3%) boys) with OSA were reviewed. Their median age, body mass index (BMI) z-score, and apnea-hypopnea index were 6.9 years, 0.87, and 9.5 events/h, respectively. The cohort was divided into three subgroups: ‘preschoolers’ (≥2 and <6 years), ‘school-age children’ (≥6 and <10 years), and ‘adolescents (≥10 and <18 years)’. The percentage of the male sex, BMI z-score, neck circumference, systolic blood pressure z-score, neutrophil-to-lymphocyte ratio, and platelet-to-lymphocyte ratio tended to increase with age. Adenoid grades tended to decrease with age. Overall, disease severity was independently correlated with neck circumference, tonsil size, and adenoid grade. Increased neck circumference and tonsillar hypertrophy were the most influential factors for younger children, whereas adenoidal hypertrophy became more important at an older age. In conclusion, gender prevalence ratio, anthropometric measures, and clinical features varied with age, and the pathogenic drivers were not necessarily the same as the aggravating ones.

## 1. Introduction

Obstructive sleep apnea (OSA) is a chronic disorder characterized by intermittent partial or complete airway obstructions during sleep with repetitive apneas and hypopneas. The impact of OSA on adults is increasing worldwide due to an increasing prevalence and a wide range of associated comorbidities, and there are also growing concerns over the prevalence of pediatric OSA [1,2]. The prevalence of pediatric OSA has been estimated to be 1.2–5.7% [3,4], and as high as 9.1% in some selected clinical populations [5]. This condition is particularly important because it is linked to adverse health outcomes, including failure to thrive, neurocognitive impairment, behavioral problems, metabolic dysregulation, and cardiovascular risks [3,6]. Early interventions are warranted to mitigate the negative effects of long-term physiological alterations, such as sleep fragmentation and hypoxia [6].

Evidence shows that adult OSA is a complex disorder with considerable diversity of patients in their pathophysiological risk factors and presenting symptoms [7,8]. These diversities are of great clinical importance, because patients with different phenotypes also have different treatment responses and health outcomes [7,8,9,10]. While there is a growing volume of studies on the further phenotyping and clustering of adult OSA patients, a similar effort for children with OSA is still relatively rare. Obesity and adenotonsillar hypertrophy are well documented risk factors for pediatric OSA [5,11,12]; however, only a few studies have focused on patient subgroups based on characteristics such as age, sex, and symptoms or elucidated the heterogeneity in their pathogeneses and treatment options [13,14,15,16,17,18].

There have been some interesting findings reported in the literature. Kang et al. demonstrated that the influence of adenoid size on pediatric OSA decreased in adolescents [13]. However, Su and colleagues reported differential risk factors for preschool and school-age children [14]. In addition, Brockmann proposed that gender bias may exist in the screening and diagnosis of pediatric OSA and suggested that further investigations on gender-specific pathophysiological mechanisms are necessary [15]. Three other studies mentioned that obese and non-obese pediatric OSA patients differed in the development, exacerbation, and resolution of the disease [16,17,18].

We hypothesized that pediatric OSA is a heterogeneous disease, in which the risk factors and clinical features vary with age among pediatric patients. This study aimed to investigate the demographics, anthropometric measurements, inflammatory markers, and disease severity parameters among different age subgroups of children with OSA.

## 2. Materials and Methods

### 2.1. Study Design and Data Collection

This was a retrospective case series. Data were retrospectively retrieved from a medical chart review of pediatric OSA patients at the Department of Otorhinolaryngology, Head and Neck Surgery, Linkou Chang Gung Memorial Hospital, Taoyuan, Taiwan between 01 March 2010–31 January 2019. The study was approved by the Institutional Review Board of the Chang Gung Memorial Foundation (No. 202000873B0). The requirement for written informed consent was waived.

### 2.2. Patient Selection and Grouping

The inclusion criteria were: (a) age 2–18 years; (b) snoring, witnessed sleep breathing pauses, nasal obstruction, mouth breathing, elevated blood pressure (BP), daytime sleepiness, learning problems, growth failure, or enuresis [19,20,21]; and (c) having complete blood count test and polysomnography at baseline. The exclusion criteria were: (a) craniofacial or neuromuscular disorders; and (b) chronic inflammatory disorders such as asthma, allergies, eczema, or other atopic/autoimmune disease [22].

The overall cohort was further divided into three subgroups based on their age: ‘preschoolers’ (≥2 years and <6 years) [23], ‘school-age children’ (≥6 years and <10 years), and ‘adolescents’ (≥10 years and <18 years) [24].

### 2.3. Measurements

#### 2.3.1. Anthropometrics

Body height, body weight, and neck circumference (NC) at the level of the thyroid cartilage were measured [25]. A body mass index (BMI) z-score was calculated for each child.

#### 2.3.2. Blood Pressure

Nocturnal BP was measured three times with a standard sphygmomanometer (Dinamap ProCare 100; GE Medical Systems Information Technologies, Inc., Milwaukee, WI, United States) between 10:00 and 11:00 PM, before the polysomnography exam. The procedure of measuring BP has been described in detail elsewhere [26]. If a high BP was noted, repeated measurements using the auscultatory method were performed. Age, sex, and height-corrected systolic BP (SBP) z-score and diastolic BP (DBP) z-score were calculated for each child [27].

#### 2.3.3. Tonsil Size and Adenoid Size

The tonsils were inspected and recorded with a size scale from 1–4 (1—tonsils within the tonsillar; 2—tonsils visible outside the anterior pillars; 3—tonsils extending three-quarters of the way to the midline; 4—tonsils meeting at the midline) [28]. 

Adenoids were scored with an adenoid grade scale from 1–4 (1—adenoids had no contact with torus tubarius; 2—the adenoids were in contact with the torus tubarius; 3—the adenoids were in contact with the torus tubarius and vomer; 4—the adenoids were in contact with the torus tubarius, vomer, and soft palate) [29].

#### 2.3.4. Systemic Inflammatory Markers

A morning blood sample was collected from each patient. If the patient had an acute systemic inflammatory condition, the blood tests were postponed until it subsided [30]. Routine platelet count and percentages of neutrophils and lymphocytes were obtained, from which platelet-to-lymphocyte ratio (PLR) and neutrophil-to-lymphocyte ratio (NLR) were calculated.

#### 2.3.5. Polysomnography

Each participant received a standard full-night in-laboratory polysomnography with a family member present to document the OSA parameters [21]. The apnea hypopnea index (AHI) was defined as the sum of all obstructive and mixed apneas (≥90% decrease in airflow for a duration of ≥2 breaths), plus hypopneas (≥50% decrease in airflow and either ≥3% desaturation or electroencephalographic arousal, for a duration of ≥2 breaths), divided by the number of hours of total sleep time, according to the 2012 American Academy of Sleep Medicine Scoring Manual [30]. Polysomnography had been re-reviewed, and then the AHI, apnea index (AI), mean oxygen saturation measured by pulse oximetry (SpO_2_), and minimal SpO_2_ were recorded for further comparisons.

### 2.4. Statistical Analysis

Most of the distributions of the variables were non-normal, assessed by using the Kolmogorov–Smirnov test. Therefore, medians and interquartile ranges were used to summarize continuous variables, and numbers with percentages were used to present categorical variables. Mann–Whitney *U* tests and Kruskal–Wallis one-way analysis of variance tests with pairwise comparisons were used to compare continuous variables, and chi-square tests were used to compare categorical variables in different groups, as appropriate. Spearman’s correlation test was used to analyze associations between variables of interest. Continuous variables were dichotomized using the median split and were analyzed for multivariate logistic regression models.

All p-values were two-sided and statistical significance was accepted at *p* < 0.05. All statistical analyses were performed using SPSS software (version 25; International Business Machines Corp., Armonk, NY, USA).

## 3. Results

### 3.1. Demographic and Clinical Characteristics of the Three Subgroups Stratified by Age 

A total of 253 Taiwanese children with OSA (70 (27.7%) girls and 183 (72.3%) boys) with a median age of 6.9 (interquartile range: 5.5–9.5) years, median BMI z-score of 0.87 (interquartile range: −0.23–1.98), and median AHI of 9.5 (interquartile range, 3.9–21.5) events/h were enrolled.

There were significant differences in age, male gender percentage, BMI z-score, NC, and adenoid grade found; however, there was no statistically significant difference in tonsil size. The preschoolers had a lower male percentage (compared to school-age children and adolescents), lower BMI (compared to school-age children and adolescents), lower NC (compared to school-age children and adolescents) and higher adenoid grade (compared to adolescents). The school-age children had a higher BMI (compared to preschoolers) and intermediate NC (compared to preschoolers and adolescents). The adolescents had a higher BMI and lower adenoid grade (compared preschoolers) and higher NC (compared to preschoolers and school-age children).

There were significant differences in SBP and DBP z-scores across subgroups. The preschoolers had a lower SBP z-score (compared to adolescents) and higher DBP z-score (compared to school-age children and adolescents). The school-age children had a lower DBP z-score (compared to the preschoolers), and the adolescents had a higher SBP z-score and lower DBP z-score (compared to the preschoolers).

There were significant differences in NLR and PLR. The preschoolers had a lower NLR (compared to the school-age children and adolescents), the school-age children had an intermediate NLR (compared to the preschoolers and adolescents), and the adolescents had a higher NLR (compared to the preschoolers and school-age children). The preschoolers had a lower PLR (compared to the adolescents).

There was a significant difference in mean SpO_2_ across subgroups. The school-age children had a higher mean SpO_2_ (compared to the adolescents). There were no statistically significant differences in AHI, AI, and minimal SpO_2_ across the subgroups.

Among all the 253 children in this study cohort, 189 (74.7%) were treated surgically. The numbers of patients who had surgical treatment were 70 (83.3%), 79 (69.9%), and 40 (71.4%) in preschoolers, school-age children, and adolescents, respectively. The *p*-value of the chi-square test was 0.08.

Among all those who received surgical treatment, 180 (95.2%) underwent adenotonsillectomy, and 9 (4.8%) received tonsillectomy. The numbers of patients with adenotonsillectomy were 68 (97.1%), 76 (96.2%), and 36 (90.0%) in preschoolers, school-age children, and adolescents, respectively. The *p*-value of the chi-square test was 0.21 (Table 1).

### 3.2. Associations between Patients’ Characteristics, Blood Pressure, Inflammatory Biomarkers, and Polysomnography Parameters in the Overall Cohort and Three Subgroups 

In the overall cohort, AHI was positively associated with BMI z-score, NC, tonsil size, adenoid grade, and DBP z-score. AI was positively associated with BMI z-score, NC, tonsil size, adenoid grade, SBP z-score, and DBP z-score. Mean SpO_2_ was inversely associated with age, BMI z-score, NC, DBP z-score, and NLR, whereas minimal SpO_2_ was inversely associated with BMI z-score, NC, tonsil size, adenoid grade, SBP z-score, and DBP z-score.

In the preschoolers, the positive associations between AHI, tonsil size, and adenoid grade were significant. Otherwise, there were no statistically significant associations between the AI, mean SpO_2_, minimal SpO_2_, and other variables of interest.

In the school-age children, AHI was positively correlated with BMI z-score, NC, tonsil size, SBP z-score, and DBP z-score. AI was positively associated with BMI z-score, NC, and DBP z-score. Mean SpO_2_ was inversely associated with BMI z-score, NC, and DBP z-score, whereas minimal SpO_2_ was inversely associated with BMI z-score, NC, SBP z-score, DBP z-score, and NLR.

In the adolescents, AHI was positively associated with BMI z-score, NC, and adenoid grade. Mean SpO_2_ was inversely associated with adenoid grade, whereas minimal SpO_2_ was inversely associated with BMI z-score, NC, and adenoid grade (Table 2).

### 3.3. Variables Independently Associated with AHI, AI, Mean SpO_2_, or Minimal SpO_2_ in the Overall Cohort Using Multivariate Logistic Regression Analysis (Figure 1)

Figure 1 shows variables independently associated with disease severity parameters in the overall cohort. NC >28.9 cm (*p* = 0.001), tonsil size >3 (*p* = 0.008), and adenoid grade >3 (*p* = 0.002) were independently correlated with AHI >9.5 events/h (Figure 1a). NC >28.9 cm (*p* = 0.001) and tonsil size >3 (*p* = 0.044) were independently correlated with AI >2.7 events/h (Figure 1b). NC >28.9 cm (*p* < 0.001), tonsil size >3 (*p* = 0.034), and DBP z-score >0.59 (*p* = 0.009) were independently correlated with mean SpO_2_ <97% (Figure 1c). BMI z-score >0.86 (*p* < 0.001), tonsil size >3 (*p* = 0.007), and DBP z-score >0.59 (*p* = 0.023) were independently correlated with minimal SpO_2_ <88% (Figure 1d).

### 3.4. Independent Variables Associated with AHI, AI, Mean SpO_2_, or Minimal SpO_2_ in the Subgroups Using Multivariate Logistic Regression Analysis (Figure 1)

Figure 2 shows variables independently associated with disease severity parameters in the subgroups.

In the preschoolers, NC >26.3 cm (*p* = 0.030) and tonsil size >3 (*p* = 0.015) were independently correlated with AHI >11.2 events/h (Figure 2a). Age >5.0 years (*p* = 0.007) was independently correlated with AI >3.6 vents/h (Figure 2b). SBP z-score >0.25 (*p* = 0.003) was independently associated with mean SpO_2_ < 97% (Figure 2c). SBP z-score >0.25 (*p* = 0.022) and DBP z-score >0.81 (*p* = 0.045) were independently correlated with minimal SpO_2_ < 85% (Figure 2d).

In the school-age children, BMI z-score >1.17 (*p* = 0.001) and tonsil size >3 (*p* = 0.015) were independently correlated with AHI >8.7 events/h (Figure 2e). BMI z-score >1.17 (*p* = 0.014), tonsil size >3 (*p* = 0.040), and DBP z-score >0.38 (*p* = 0.017) were independently correlated with AI >2.8 events/h (Figure 2f). NC >29.8 cm (*p* = 0.009) was independently associated with mean SpO_2_ < 97% (Figure 2g). BMI z-score >1.17 (*p* < 0.001) was independently correlated with minimal SpO_2_ < 88% (Figure 2h). 

In the adolescents, adenoid grade >3 (*p* = 0.007) was independently associated with AHI >9.8 events/h (Figure 2i). Adenoid grade >3 (*p* = 0.007) were independently correlated with AI >2.2 events/h (Figure 2j). Tonsil size >3 (*p* = 0.024) and adenoid grade >3 (*p* = 0.024) were independently correlated with mean SpO_2_ < 96% (Figure 2k). Tonsil size >3 (*p* = 0.037) and adenoid grade >3 (*p* = 0.021) were independently associated with minimal SpO_2_ < 87% (Figure 2l).

## 4. Discussion

In this study, we investigated how various factors differed among age subgroups and how they were related to OSA severity. The three subgroups were similar in disease severity, which enabled us to observe inter-group differences in pathogenic factors. Correlations between independent variables and disease severity parameters demonstrated how they might have contributed as aggravating factors. This effort was important for understanding how the development and progression of pediatric OSA changes with age. In the discussion, we will address our findings sequentially on domains of factors including gender, anthropometric measures, blood pressure, and inflammatory biomarkers.

First, the male percentage increased across subgroups from the preschoolers to adolescents. OSA is traditionally considered to be a male disease in adults, with male–female ratios ranging from 3:1 to 10:1 [31]. However, this difference in the prevalence between genders is much more tenuous in pediatric patients [15]. One of the proposed explanations is the gender differences in upper airway anatomy and respiratory physiology influenced by sex hormones [31,32]. However, in this study, the significant incremental increase in male percentage from preschool to school to adolescent age groups indicated that this gender discrepancy existed and aggravated even before puberty. Further studies are warranted to investigate sex hormone-independent underlying mechanisms of the gender differences in pediatric OSA. On the other hand, the correlation between disease severity and gender was nonsignificant in both the overall cohort and all subgroups, suggesting that male sex may be a contributing factor to the presence but not the severity of the disease.

Second, there were several notable findings regarding anthropometric measures. BMI z-score increased across subgroups, suggesting that weight status played a larger role at an older age. This observation is consistent with the fact that obesity is known to be a stronger risk factor for OSA in adults than in children [14]. NC exhibited a similar trend; however, it would not be proper to make a similar inference since the increase in NC could be a result of normal growth, and there is currently no well-validated growth chart reference for pediatric NC as there is for pediatric BMI. On the other hand, the adenoid grade was higher in the preschoolers, suggesting that the enlargement of adenoid was a more important pathogenic factor in younger pediatric OSA patients. Interestingly, from the correlation analysis, we could see that disease aggravating factors were different among subgroups. The size of the tonsil and adenoid were the two dominant variables contributing to disease severity for preschoolers. For school-age students, weight status and neck circumference were more influential. For adolescents, the effects on disease severity were more universally distributed, suggesting that the exacerbation of OSA might be more multi-factorial in this population. Pathophysiological factors not only vary with age, but those factors involved in acquiring pediatric OSA may also be different from those involved in worsening the disease. Future studies are warranted to better understand the underlying mechanisms.

BP also demonstrated interesting patterns across subgroups. SBP z-score was highest in the adolescents, which may be a reflection of a higher BMI z-score based on the known association between higher BMI and SBP [33,34,35]. In contrast, DBP z-score was highest in the preschool-age children. DBP has been shown to reflect arterial resistance, and the elevation of DBP in young individuals is usually suggestive of comorbidities [36,37,38]. We previously demonstrated that non-obese pediatric OSA patients had a higher level of systemic inflammation than obese pediatric OSA patients [17], which may explain why the preschool-age children, the least obese subgroup in the study, had the highest DBP z-score among all.

NLR and PLR are recently identified biomarkers for systemic inflammation. The literature has reported their linkage to higher disease severity and poorer outcomes in some cancers and autoimmune disorders [39,40,41]. PLR has also been found to be an independent predictor for cardiovascular diseases in OSA by Handan et al [42]. In our results, except for an inverse association between NLR and saturation in two subgroups, no other significant association was observed between these two inflammatory biomarkers and disease severity. We postulate that the inflammatory stress caused by OSA exacerbates with time. Therefore, for pediatric OSA patients, in whom the disease incubation time is not as long as that of typical middle-aged adult patients, the presence of systemic inflammation is more insidious and difficult to monitor. This inference can be also supported by the observation that the levels of both NLR and PLR were both significantly higher for older subgroups.

Concordant with these phenotypic differences, we found that the risk factors of the three age subgroups were intrinsically distinct. As adenotonsillectomy and anti-inflammation therapies are both important treatments for pediatric OSA, every child patient must be treated with individually tailored plans. Not surprisingly, NC, tonsil size, and adenoid grade were associated with OSA severity in general, consistent with previous reports. However, our findings show that, while adenotonsillar hypertrophy is still the most distinguishable and well-documented risk factor for children with OSA, different variables seem to contribute to disease severity to different extents across age subgroups. For preschoolers, NC and tonsil size were most dominant for high AHI, and SBP and DBP were most dominant for low SpO_2_; the influence of BMI z-score on high AHI and AI and low minimal SpO_2_ started to emerge in school-age children; for adolescents, adenoidal hypertrophy was the most dominant for high AHI and AI, and adenotonsillar hypertrophy was the most dominant for low mean and minimal SpO_2_ and should be monitored and managed.

The main contribution of this study is that it highlights the possible pathogenic drivers and aggravating factors of pediatric OSA among various age subgroups. The limitations of the study include the single ethnicity of the subjects, which may limit the generalizability of the results. Also, since the data of this study were retrieved from a medical chart review, variables of interest were limited only to the information collected from the clinical service provided at that time. Therefore, some promising systemic inflammatory biomarkers, such as high-sensitivity C-reactive protein and cytokines, were not be included in this clinical investigation. Otherwise, this study was a retrospective case series and therefore we could not conclude causative relationships. Future studies are warranted to better understand the underlying mechanisms.

## 5. Conclusions

Gender prevalence ratio, anthropometric measures, and clinical features were intrinsically distinct among the preschoolers, school-age children, and adolescents with OSA in this study. The pathogenic drivers were not necessarily the same as the aggravating ones. Clinical management should be tailored individually. Future research on age-related underlying mechanisms of pediatric OSA is warranted to aid the development of age-specific precision treatment.

## Figures and Tables

**Figure 1 ijerph-17-04663-f001:**
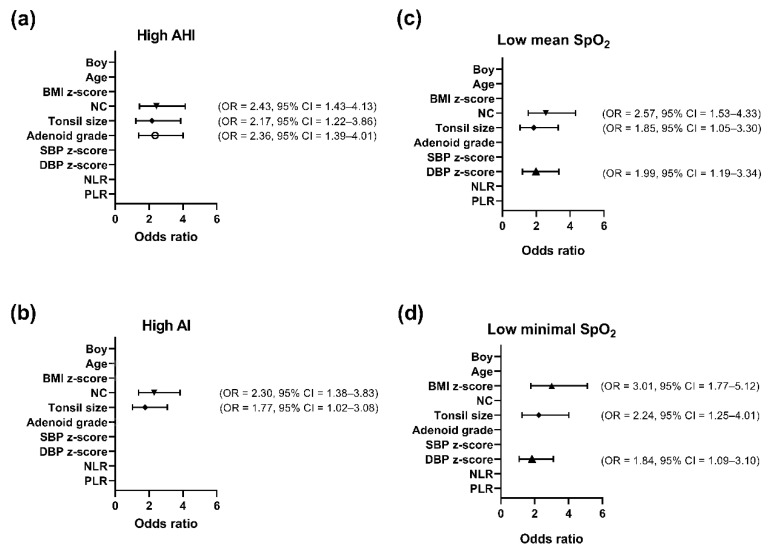
Variables independently associated with AHI, AI, mean SpO2, and minimal SpO2 in the overall cohort. (**a**) High NC, tonsil size, and adenoid grade were independently associated with high AHI.; (**b**) high NC and tonsil size were independently associated with high AI.; (**c**) high NC, tonsil size and DBP z-score were independently associated with low mean SpO2.; (**d**) high BMI z-score, tonsil size, and DBP z-score were independently associated with low minimal SpO2. Abbreviations: AHI—apnea–hypopnea index; AI—apnea index; BMI—body mass index; CI—confidence interval; OR—odds ratio; NC—neck circumference; NLR—neutrophil-to-lymphocyte ratio; PLR—platelet-to-lymphocyte ratio; SBP—systolic blood pressure; DBP—diastolic blood pressure; SpO2—oxygen saturation measured by pulse oximetry.

**Figure 2 ijerph-17-04663-f002:**
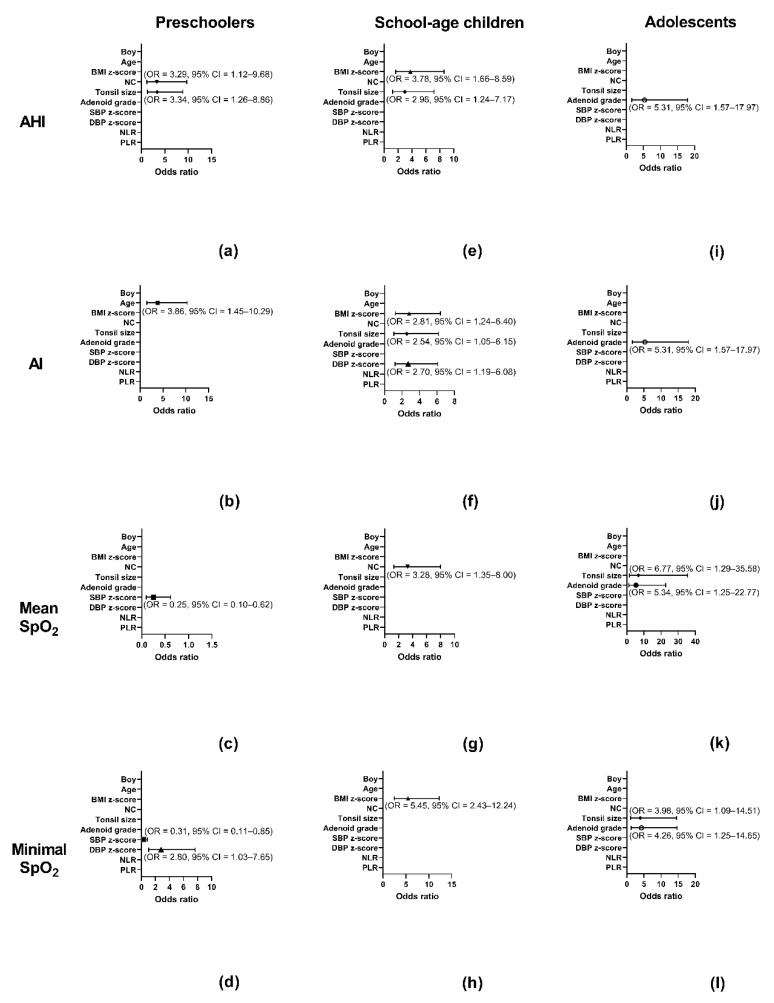
Variables (odds ratios and 95% confidence intervals) independently associated with AHI, AI, mean SpO_2_, and minimal SpO_2_ in various age subgroups. In the preschoolers—(**a**) high NC and tonsil size independently correlated with high AHI; (**b**) high age was independently correlated with high AI; (**c**) high SBP z-score was independently associated with low mean SpO_2_; and (**d**) high SBP z-score and high DBP z-score were independently associated with low minimal SpO_2_. In the school-age children—(**e**) high BMI z-score, tonsil size, and DBP z-score were independently correlated with high AHI; (**f**) high NC and adenoid grade were independently correlated with high AI; (**g**) high NC was independently associated with low mean SpO_2_; and (**h**) high BMI z-score was independently correlated with low minimal SpO_2_. In the adolescents—(**i**) high adenoid grade was independently associated with high AHI; (**j**) high adenoid grade was independently correlated with high AI; (**k**) high tonsil size and adenoid grade were independently correlated with low mean SpO_2_; and (**l**) high tonsil size and adenoid grade were independently associated with low minimal SpO_2_. Abbreviations: AHI—apnea–hypopnea index; AI—apnea index; BMI—body mass index; CI—confidence interval; OR—odds ratio; NC—neck circumference; NLR—neutrophil-to-lymphocyte ratio; PLR—platelet-to-lymphocyte ratio; SBP—systolic blood pressure; DBP—diastolic blood pressure; SpO_2_—oxygen saturation measured by pulse oximetry.

**Table 1 ijerph-17-04663-t001:** Demographic and clinical characteristics of the three subgroups stratified by age.

Variables	Preschoolers	School-age children	Adolescents	*p*-Value^1^
Patients	*n* = 84	*n* = 113	*n* = 56	
Boys (*n*)	**51 (60.7)**	88 (77.9)	**44 (78.6)**	**0.014**
Age (years)	**5.0 (4.1–5.5) ^2^**	**7.0 (6.3–8.5) ^2,3^**	**11.0 (11.2–12.0) ^2,3^**	**<0.001**
BMI (kg/m^2^) z-score	**0.24 (-0.55–1.01) ^2^**	**1.17 (−0.07–2.05) ^2^**	**1.41 (0.69–2.05) ^2^**	**<0.001**
NC (cm)	**26.3 (25.0–2.05) ^2^**	**29.8 (26.4–31.8) ^2,3^**	**32.3 (30.8–35.0) ^2,3^**	**<0.001**
Tonsil size	3 (3–4)	3 (3–4)	3 (3–4)	0.725
Adenoid grade	**4 (3–4) ^2^**	3 (3–4)	**3 (2–4) ^2^**	**0.026**
SBP (mmHg) z-score	**0.25 (−0.56–1.45) ^2^**	0.45 (−0.54–1.80)	**0.85 (0.02–1.97) ^2^**	**0.046**
DBP (mmHg) z-score	**0.81 (0.30–1.49) ^2^**	**0.38 (−0.11–1.05) ^2^**	**0.42 (−0.10–1.05) ^2^**	**0.006**
NLR	**1.06 (0.82–1.53) ^2^**	**1.24 (0.92–1.64) ^3^**	**1.51 (1.07–2.01) ^2,3^**	**0.002**
PLR	**94.1 (79.2–117.2) ^2^**	103.3 (86.5–129.5)	**110.2 (93.5–127.0) ^2^**	**0.024**
AHI (events/h)	11.2 (5.2–22.4)	8.7 (3.6–20.4)	9.8 (3.6–22.4)	0.667
AI (events/h)	3.6 (1.5–8.1)	2.8 (1.2–8.7)	2.2 (1.2–9.5)	0.549
Mean SpO_2_ (%)	97 (95–98)	**97 (95–98) ^3^**	**96 (94–97) ^3^**	**0.038**
Minimal SpO_2_ (%)	85 (81–91)	88 (83–92)	87 (82–91)	0.156

Note: Data are summarized as median (interquartile range) or *n* (%) as appropriate. Abbreviations: AHI—apnea–hypopnea index; AI—apnea index; BMI—body mass index; DBP—diastolic blood pressure; NC—neck circumference; NLR—neutrophil-to-lymphocyte ratio; PLR—platelet-to-lymphocyte ratio; SBP—systolic blood pressure; SpO_2_—oxygen saturation measured by pulse oximetry. ^1^—Data were compared using Kruskal–Wallis tests with pairwise comparisons for continuous variables, and the chi-square test for categorical variables. ^2^—*p*-value <0.05 when the variable in the preschoolers was compared with the school-age children or adolescents.; ^3^—*p*-value <0.05 when the variable in the school-age children was compared with the adolescents; significant *p*-values are marked in bold.

**Table 2 ijerph-17-04663-t002:** Associations between patients’ characteristics, blood pressure, inflammatory biomarkers, and polysomnography parameters in the overall cohort and three subgroups.

Variables	Boys	Age	BMI z-Score	NC	Tonsil Size	Adenoid Grade	SBP z-Score	DBP z-Score	NLR	PLR
Overall Group (*n* = 253)										
AHI (events/h)	0.05 (0.454)	−0.01 (0.983)	**0.25 (<0.001)**	**0.22 (<0.001)**	**0.23 (<0.001)**	**0.28 (<0.001)**	0.09 (0.154)	**0.17 (0.007)**	0.10 (0.127)	−0.05 (0.413)
AI (events/h)	0.01 (0.989)	−0.01 (0.975)	**0.19 (0.002)**	**0.17 (0.006)**	**0.16 (0.012)**	**0.23 (<0.001)**	**0.04 (0.530)**	**0.12 (0.049)**	0.09 (0.152)	−0.03 (0.694)
Mean SpO_2_ (%)	0.01 (0.989)	**−0.14 (0.023)**	**−0.24 (<0.001)**	**−0.23 (<0.001)**	−0.04 (0.511)	−0.09 (0.175)	−0.01 (0.866)	**−0.20 (0.001)**	**−0.15 (0.017)**	−0.01 (0.859)
Minimal SpO_2_ (%)	−0.04 (0.518)	0.04 (0.582)	**−0.29 (<0.001)**	**−0.26 (<0.001)**	**−0.17 (0.008)**	**−0.17 (0.006)**	−0.07 (0.268)	**−0.19 (0.003)**	−0.10 (0.111)	−0.01 (0.972)
Preschoolers (*n* = 84)										
AHI (events/h)	0.18 (0.099)	0.12 (0.284)	0.15 (0.174)	0.20 (0.070)	**0.34 (0.002)**	**0.25 (0.023)**	−0.07 (0.524)	0.08 (0.469)	0.02 (0.882)	0.03 (0.785)
AI (events/h)	0.11 (0.314)	0.16 (0.138)	0.12 (0.298)	0.10 (0.360)	0.21 (0.057)	0.10 (0.360)	−0.14 (0.212)	0.03 (0.799)	0.04 (0.724)	0.08 (0.446)
Mean SpO_2_ (%)	−0.09 (0.400)	−0.3 (0.798)	−0.10 (0.377)	−0.09 (0.399)	−0.04 (0.728)	−0.02 (0.896)	0.21 (0.059)	−0.10 (0.354)	−0.02 (0.874)	−0.17 (0.123)
Minimal SpO_2_ (%)	−0.04 (0.726)	0.13 (0.249)	−0.18 (0.101)	−0.21 (0.058)	−0.19 (0.085)	−0.09 (0.422)	0.10 (0.358)	−0.17 (0.131)	0.03 (0.802)	−0.20 (0.067)
School−Age Children (*n* = 113)										
AHI (events/h)	−0.08 (0.376)	0.08 (0.374)	**0.31 (0.001)**	**0.30 (0.001)**	**0.20 (0.037)**	0.18 (0.063)	**0.23 (0.016)**	**0.27 (0.004)**	0.07 (0.476)	−0.07 (0.473)
AI (events/h)	−0.09 (0.326)	0.10 (0.295)	**0.25 (0.008)**	**0.29 (0.002)**	0.13 (0.169)	0.16 (0.088)	0.15 (0.103)	**0.20 (0.039)**	0.10 (0.302)	−0.03 (0.726)
Mean SpO_2_ (%)	0.07 (0.496)	−0.16 (0.101)	**−0.31 (<0.001)**	**−0.29 (0.002)**	−0.03 (0.757)	−0.07 (0.486)	−0.09 (0.344)	**−0.31 (0.001)**	−0.18 (0.054)	0.07 (0.480)
Minimal SpO_2_ (%)	0.02 (0.844)	−0.18 (0.059)	**−0.43 (<0.001)**	**−0.43 (<0.001)**	−0.15 (0.112)	−0.09 (0.344)	**−0.20 (0.039)**	**−0.24 (0.012)**	**−0.19 (0.048)**	0.07 (0.447)
Adolescents (*n* = 56)										
AHI (events/h)	0.21 (0.130)	0.17 (0.21)	**0.37 (0.005)**	**0.34 (0.010)**	0.18 (0.186)	**4.0 (0.002)**	0.07 (0.630)	0.02 (0.860)	0.34 (0.010)	0.01 (0.956)
AI (events/h)	0.07 (0.615)	0.23 (0.085)	0.24 (0.074)	0.22 (0.104)	0.16 (0.242)	**0.41 (0.002)**	0.12 (0.375)	0.06 (0.645)	0.20 (0.143)	−0.04 (0.785)
Mean SpO_2_ (%)	0.04 (0.750)	−0.24 (0.081)	−0.17 (0.219)	−0.24 (0.081)	−0.13 (0.359)	**−0.29 (0.030)**	−0.06 (0.640)	−0.18 (0.180)	−0.19 (0.154)	0.04 (0.768)
Minimal SpO_2_ (%)	−0.24 (0.071)	−0.07 (0.594)	**−0.34 (0.010)**	**−0.36 (0.006)**	−0.20 (0.132)	**−0.34 (0.010)**	−0.05 (0.700)	0.01 (0.992)	−0.22 (0.102)	−0.02 (0.869)

Note: Data are summarized as correlation coefficients (*p*-values). Abbreviations: AHI—apnea–hypopnea index; AI—apnea index; BMI—body mass index; DBP—diastolic blood pressure; NC—neck circumference; NLR—neutrophil-to-lymphocyte ratio; PLR—platelet-to-lymphocyte ratio; SBP—systolic blood pressure; SpO_2_—oxygen saturation measured by pulse oximetry. Variables of interest were analyzed using Spearman’s correlation test. Significant *p*-values are marked in bold.

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
