# Peer review of "Differences in Anthropometric and Clinical Features among Preschoolers, School-Age Children, and Adolescents with Obstructive Sleep Apnea—A Hospital-Based Study in Taiwan"

_ijerph, 2020, doi:10.3390/ijerph17134663_

Round 1
Reviewer 1 Report
In this article, the authors did a descriptive analysis of OSA in the pediatric population in Taiwan. The study was done well and written meticulously.
However, I have one doubt regarding the methods section. The authors have chosen NLR and PLR as inflammatory markers. Still, the first authors have mentioned in the discussion regarding his previous work(citation 17) where Interleukins are studied as the markers. (1,2,3)
Is there any rationale for choosing NLR or PLR over the cytokines ? as we all know, adipose tissue is an endocrine organ that plays a role in chronic inflammation with cytokines and other factors.
1: DOI: 10.1183/13993003.congress-2019.PA2582
2: DOI: 10.17219/acem/47735
3: DOI: 10.3390/jcm9020579
Finally, the authors mentioned the generalisability regarding the data from this study as ethnicity plays some role in OSA's incidence.
Author Response
June 15, 2020
IJERPH
Dear Reviewer,
We appreciate reviewers’ valuable comments on the previous version of our manuscript. We have made a point-by-point response to the reviewers' comments and suggestions and marked up the changes made from the previous article file as a revised manuscript file. Please find the “response to reviewers” section attached at the end of the letter in which we address all the mentioned concerns. Careful revisions have been made in the manuscript according to the reviewers’ suggestions.
Thank you again for reviewing our paper. We look forward to your further information.
Sincerely yours,
|
Li-Ang Lee, M.D., M.Sc., F.I.C.S.
|
AUTHORS’ REPLY TO REVIEWERS’ COMMENTS:
Reviewer 1’s comments:
In this article, the authors did a descriptive analysis of OSA in the pediatric population in Taiwan. The study was done well and written meticulously.
REPLY. We thank the reviewer for these positive comments. With our enthusiasm in the research field of pediatric obesity and related comorbidities, we find your words very encouraging.
- The authors have chosen NLR and PLR as inflammatory markers. Still, the first authors have mentioned in the discussion regarding his previous work (citation 17) where Interleukins are studied as the markers. (1,2,3) (citation 17) Is there any rationale for choosing NLR or PLR over the cytokines? As we all know, adipose tissue is an endocrine organ that plays a role in chronic inflammation with cytokines and other factors.
REPLY. Thank you very much for this in-depth comment. Unfortunately, we do not have that data in our cohort.
The practice of healthcare in Taiwan is almost universally covered but then also universally and largely restricted to the national health insurance system. Complete blood counts (CBC) and differential white cell (DC) counts are routine blood tests in the work-up for pediatric OSA patients, from which we can calculate NLR and PLR. On the other hand, cytokines, which are much more costly compared to CBC/ DC, are not considered a part of standard practice. Cytokines are rarely examined, mostly in research settings, since they are not covered by the insurance.
The data of this study was retrieved from medical chart reviews, so the information we could use was limited to those obtained from the clinical service provided at that time. We’ve revised the last paragraph in our Discussion section to address this limitation.
We could include more inflammatory biomarkers in our future prospective studies though. Again, thank you for the suggestion.
Modified text, Page 13, Lines 319-329
Add:
‘NLR and PLR are recently identified biomarkers for systemic inflammation. The literature has reported their linkage to higher disease severity and poorer outcomes in some cancers and autoimmune disorders [39-41]. PLR has also found to be an independent predictor for cardiovascular diseases in OSA by Handan et al [42]. In our results, except for an inverse association between NLR and saturation in two subgroups, no any other significant association was observed between these two inflammatory biomarkers and disease severity. We postulate that the inflammatory stress caused by OSA exacerbates with time. Therefore, for pediatric OSA patients, in whom the disease incubation time is not as long as that of typical middle-aged adult patients, the presence of systemic inflammation is more insidious and more difficult to monitor. This inference can be also supported by the observation that the levels of both NLR and PLR were both going significantly higher for older subgroups.’
Modified text, Page 13, Lines 345-349
‘… include the single ethnicity of the subjects, which may limit the generalizability of the results. In addition, this was a cross-sectional investigation, and therefore we could not conclude causative …’
à
‘… include the single ethnicity of the subjects, which may limit the generalizability of the results. Also, since the data of this study were retrieved from a medical chart review, variables of interest were limited only to the information collected from the clinical service provided at that time. Therefore, some promising systemic inflammatory biomarkers, such as high-sensitivity C-reactive protein and cytokines, were not be included for this clinical investigation. Besides, this study was a retrospective case series, and therefore we could not conclude causative relationships. Future studies are …’
- Finally, the authors mentioned the generalisability regarding the data from this study as ethnicity plays some role in OSA's incidence.
REPLY. Thank you for making this point. We’ve revised the title of this study to better present the nature and also limitations of this study.
Modified text, Page 1, Lines 2-5
‘Differences in Demographic, Anthropometric, Inflammatory, and Clinical Features among Pre-school Children, School-age Children, and Adolescents with Obstructive Sleep Apnea’
à
‘Differences in Anthropometric and Clinical Features among Preschoolers, School-age Children, and Adolescents with Obstructive Sleep Apnea – A Hospital-based Study in Taiwan’
Reviewer 2 Report
In this retrospective cross-sectional observation study the authors try to specify the factors involved in obstructive sleep apnea (OSA) in Taiwanese children. The OSA represents part of a large group of pathologies of respiratory sleep disorders (RSD) with potential consequences if untreated. The paper is well written, and since many aspects of this pathology in children are still debated, the data on the variables studied add interesting information in the field.
- Title : The title is very general. As the authors mention, the limitation of the single ethnicity of the subjects may limit the generalizability of the results; thus, the words in Taiwan should be added in the title. Furthermore, the word inflammatory should propably be eliminated since the reader expects more specific data on inflammatory factors.
- Methods section: Patient selection: The authors have divided the children into three subgroups based on their age; tanner stage should be more appropriate, especially for the anthropometric data (i.e neck circumference). Under this pont of view the major differences should be found in the 10-18 years old children group; thus, subgroups groups between 10-14 years and 14-18 years would be more homogeneous Are there any differences between these subgroups?
- Methods section: Systemic Inflammatory Markers. Was sedimentation rate or CRP measured?
- Results Do the authors know what was the clinical management of the children in regards to the prevalent signs? If yes, a table with the clinical management and the related criteria should be very interesting.
- English language: minor syntax and grammatical errors
Author Response
June 15, 2020
IJERPH
Dear Reviewer,
We appreciate reviewers’ valuable comments on the previous version of our manuscript. We have made a point-by-point response to the reviewers' comments and suggestions and marked up the changes made from the previous article file as a revised manuscript file. Please find the “response to reviewers” section attached at the end of the letter in which we address all the mentioned concerns. Careful revisions have been made in the manuscript according to the reviewers’ suggestions. Please see the attachment.
Thank you again for reviewing our paper. We look forward to your further information.
Sincerely yours,
|
Li-Ang Lee, M.D., M.Sc., F.I.C.S.
|
AUTHORS’ REPLY TO REVIEWERS’ COMMENTS:
Reviewer 2’s comments:
- Title: The title is very general. As the authors mention, the limitation of the single ethnicity of the subjects may limit the generalizability of the results; thus, the words in Taiwan should be added in the title. Furthermore, the word inflammatory should probably be eliminated since the reader expects more specific data on inflammatory factors.
REPLY. Thank you very much for this valuable comment. We’ve revised the title of the study to better present the nature and limitations of our study (at line 2-5).
Modified text, Page 1, Lines 2-5
‘Differences in Demographic, Anthropometric, Inflammatory, and Clinical Features among Pre-school Children, School-age Children, and Adolescents with Obstructive Sleep Apnea’
à
‘Differences in Anthropometric and Clinical Features among Preschoolers, School-age Children, and Adolescents with Obstructive Sleep Apnea – A Hospital-based Study in Taiwan’
- Methods section: Patient selection: The authors have divided the children into three subgroups based on their age; tanner stage should be more appropriate, especially for the anthropometric data (i.e neck circumference). Under this point of view the major differences should be found in the 10-18 years old children group; thus, subgroups groups between 10-14 years and 14-18 years would be more homogeneous. Are there any differences between these subgroups?
REPLY. Thank you for this in-depth suggestion! It would be really interesting to look at the differences between these subgroups, especially if we want to understand the role of sex hormones in the development and progression of pediatric OSA. However, the sample size of the adolescent subgroup (n = 56) was not very large. We further divided them into 10-14 (n = 53) and 14-18 (n = 3) year of age and had the following result:
There were significant differences in age (median [interquartile range]) (11.0 [10.2–11.7] vs 14.8 [14.1–14.8], p < 0.001), SBP z-score (0.99 [0.19–1.99] vs -0.77 [-0.77–2.00], p = 0.015), and DBP z-score (0.44 [0.06–1.10] vs -0.28 [-0.28–1.01], p = 0.007).
We didn’t find the result of too much value to report, so we didn’t include it in the revised manuscript. Thank you for bringing up this valuable suggestion though. We will further look into it in our future studies when we’ve collected enough case numbers.
- Methods section: Systemic Inflammatory Markers. Was sedimentation rate or CRP measured?
REPLY. Unfortunately, they were not. The practice of healthcare in Taiwan is almost universally covered by the national health insurance but then also largely restricted by the regulations of the system. Complete blood counts (CBC) and differential white cell (DC) counts are routine blood tests in the work-up for pediatric OSA patients, from which we can calculate NLR and PLR. On the other hand, sedimentation rate and CRP are not considered a part of standard practice.
The data of this study was retrieved from medical chart reviews, so the information we could use was limited to those obtained from the clinical service provided at that time. We’ve revised the last paragraph in our Discussion section to address this limitation.
We could include more inflammatory biomarkers in our future prospective studies though. Again, thank you for the suggestion
Modified text, Page 13, Lines 345-349
‘… include the single ethnicity of the subjects, which may limit the generalizability of the results. In addition, this was a cross-sectional investigation, and therefore we could not conclude causative …’
à
‘… include the single ethnicity of the subjects, which may limit the generalizability of the results. Also, since the data of this study were retrieved from a medical chart review, variables of interest were limited only to the information collected from the clinical service provided at that time. Therefore, some promising systemic inflammatory biomarkers, such as high-sensitivity C-reactive protein and cytokines, were not be included for this clinical investigation. Besides, this study was a retrospective case series, and therefore we could not conclude causative relationships. Future studies are …’
- Results: Do the authors know what was the clinical management of the children in regards to the prevalent signs? If yes, a table with the clinical management and the related criteria should be very interesting.
REPLY. Yes, a few of our authors are experienced experts of pediatric OSA in Taiwan, and we are more than happy to share our clinical managements.
Among all the 253 children in this study cohort, 189 (74.7%) were treated surgically. The number (percentage) of patients with surgical treatment were 70 (83.3%), 79 (69.9%), and 40 (71.4%) in preschoolers, school-age children, and adolescents, respectively. The p value of chi-square test was 0.08.
Among all those received surgical treatment, 180 (95.2%) underwent adenotonsillectomy, 9 (4.8%) received tonsillectomy. The number (percentage) of patients with adenotonsillectomy were 68 (97.1%), 76 (96.2%), and 36 (90.0%) in preschoolers, school-age children, and adolescents, respectively. The p value of chi-square test was 0.21.
We mentioned about the clinical management and provided a few references to the readers for their information. Thank you for helping us to improve the manuscript.
Modified text, Page 4, Lines 168-175
Add
à
‘Among all the 253 children in this study cohort, 189 (74.7%) were treated surgically. The number (percentage) of patients with surgical treatment were 70 (83.3%), 79 (69.9%), and 40 (71.4%) in preschoolers, school-age children, and adolescents, respectively. The p value of chi-square test was 0.08.
Among all those received surgical treatment, 180 (95.2%) underwent adenotonsillectomy, 9 (4.8%) received tonsillectomy. The number (percentage) of patients with adenotonsillectomy were 68 (97.1%), 76 (96.2%), and 36 (90.0%) in preschoolers, school-age children, and adolescents, respectively. The p value of chi-square test was 0.21.’
- English language: minor syntax and grammatical errors
REPLY. Thank you for reminding us of the issue. We’ve sent our manuscript for English editing by a native speaker and attached the certificate in our reply. Of course, if necessary, we will send the final version for re-checking and corrections of syntax and grammatical errors, again.
Reviewer 3 Report
Report entitled „Differences in Demographic, Anthropometric, Inflammatory, and Clinical Features among Pre-school Children, School-age Children, and Adolescents with Obstructive Sleep Apnea” risk factors and clinical features vary with age among pediatric patients. The study has potentially interesting results. However, statistical analysis and presentation of results has to be improved to properly assess the study.
The study design is unclear. First authors state that this is a cross-sectional observational study, and later state that the data were retrospectively retrieved which is contradictory. This has to correct in the manuscript.
The statistical analysis section does not state if the normal distribution of variables was inspected. Further states only test for normal distribution. Therefore, results obtained could differ from obtained. The analysis should be redone and better described. Additionally, correction for multiple comparisons should be incorporated. Without the aforementioned, it is hard to analyze results and discussion.
Section 3.3 and 3.4 is extremely hard to read. It would be greatly advisable to present results as a table instead of the text and figure as it would be more transparent.
The style of the manuscript should be improved: in many places’ repetitions occur, multiple/singular form is incorrectly used.
In the abstract, no aim of the study is described.
Author Response
Please see the attachment.
June 15, 2020
IJERPH
Dear Reviewer,
We appreciate reviewers’ valuable comments on the previous version of our manuscript. We have made a point-by-point response to the reviewers' comments and suggestions and marked up the changes made from the previous article file as a revised manuscript file. Please find the “response to reviewers” section attached at the end of the letter in which we address all the mentioned concerns. Careful revisions have been made in the manuscript according to the reviewers’ suggestions.
Thank you again for reviewing our paper. We look forward to your further information.
Sincerely yours,
|
Li-Ang Lee, M.D., M.Sc., F.I.C.S.
|
AUTHORS’ REPLY TO REVIEWERS’ COMMENTS:
Reviewer 3’s comments:
- The study design is unclear. First authors state that this is a cross-sectional observational study, and later state that the data were retrospectively retrieved which is contradictory. This has to correct in the manuscript.
REPLY. Thank you very much for this valuable comment. The study should be defined as a retrospective case series investigation. We are sorry for the inaccurate and misleading presentation. We’ve corrected it in the Abstract, the first paragraph of Method, and the last paragraph of Discussion section.
Modified text, Page 1, Lines 33-34
‘… This cross-sectional observational study retrospectively reviewed 253 …’
à
‘… This retrospective study aims to investigate age-related differences in anthropometric and clinical features of this population. A total of 253 …’
- The statistical analysis section does not state if the normal distribution of variables was inspected. Further states only test for normal distribution. Therefore, results obtained could differ from obtained. The analysis should be redone and better described. Additionally, correction for multiple comparisons should be incorporated. Without the aforementioned, it is hard to analyze results and discussion.
REPLY. Thank you for this in-depth suggestion. We’ve redone all the statistical analysis and revised our manuscript extensively. Different analytical methods were applied, and variables were presented in different ways carefully. Please find our changes in:
- Abstract: results and conclusion revised accordingly
- Method: 2.4 Statistical Analysis
- Results: 3.1 ~ 3.4
- Discussion: the 1st, 3rd, 5th, and 6th paragraphs for major changes
We hope that the revised manuscript satisfactorily meets your expectations.
Modified text, Pages 1-2, Lines 36-46
Amended:
‘… Their median age, body mass index (BMI) z-score, and apnea-hypopnea index were 6.9 years, 0.87, and 9.5 events/h, respectively. The cohort was divided into three subgroups: ‘preschoolers (≥ 2 and < 6 years)’, ‘school-age children (≥ 6 and < 10 years)’, and ‘adolescents (≥ 10 and < 18 years)’. The percentage of the male sex, BMI z-score, neck circumference, systolic blood pressure z-score, neutrophil-to-lymphocyte ratio, and platelet-to-lymphocyte ratio tended to increase with age. Adenoid grades tended to decrease with age. Overall, disease severity was independently correlated with neck circumference, tonsil size, and adenoid grade. Increased neck circumference and tonsillar hypertrophy were the most influential factors for younger children, whereas adenoidal hypertrophy became more important at an older age. In conclusion, gender prevalence ratio, anthropometric measures, and clinical features varied with age, and the pathogenic drivers were not necessarily the same as the aggravating ones.’
Modified text, Pages 3-4, Lines 131-138
Amended:
‘Most of the distributions of the variables were non-normal, assessed by using the Kolmogorov-Smirnov test. Therefore, medians and interquartile ranges were used to summarize continuous variables, and numbers with percentages were used to present categorical variables. Mann-Whitney U tests and Kruskal-Wallis one-way analysis of variance tests with pairwise comparisons were used to compare continuous variables, and chi-square tests were used to compare categorical variables in different groups, as appropriate. Spearman’s correlation test was used to analyze associations between variables of interest. Continuous variables were dichotomized using the median split and were analyzed for multivariate logistic regression models. …’
Modified text, Page 4, Lines 144-146
Amended:
‘A total of 253 consecutive Taiwanese children with OSA (70 [27.7%] girls and 183 [72.3%] boys) with a median age of 6.9 (interquartile range, 5.5–9.5) years, median BMI z-score of 0.87 (interquartile range, -0.23–1.98), and median AHI of 9.5 (interquartile range, 3.9–21.5) events/h were enrolled.’
Modified text, Page 4, Line 153
Amended:
‘… adolescents had higher BMI and lower adenoid grade (compared preschoolers) and higher NC …’
Modified text, Pages 4, Lines 160-175
Amended:
‘There were significant differences in NLR and PLR. The preschoolers had a lower NLR compared to the school-age children and adolescents, the school-age children had an intermediate NLR compared to the preschoolers and adolescents, and the adolescents had a higher NLR compared to the preschoolers and school-age children. The preschoolers had a lower PLR compared to the adolescents.
There was a significant difference in mean SpO2 across subgroups. The school-age children had a higher mean SpO2 compared to the adolescents. There were no statistically significant differences in AHI, AI, and minimal SpO2 across subgroups.
Among all the 253 children in this study cohort, 189 (74.7%) were treated surgically. The number (percentage) of patients with surgical treatment were 70 (83.3%), 79 (69.9%), and 40 (71.4%) in preschoolers, school-age children, and adolescents, respectively. The p value of chi-square test was 0.08.
Among all those received surgical treatment, 180 (95.2%) underwent adenotonsillectomy, 9 (4.8%) received tonsillectomy. The number (percentage) of patients with adenotonsillectomy were 68 (97.1%), 76 (96.2%), and 36 (90.0%) in preschoolers, school-age children, and adolescents, respectively. The p value of chi-square test was 0.21.’
Modified text, Pages 4-5, Table 1
Amended data.
Modified text, Pages 5-6, Lines 189-203
Amended:
‘In the overall cohort, AHI was positively associated with BMI z-score, NC, tonsil size, adenoid grade and DBP z-score. AI was positively associated with BMI z-score, NC, tonsil size, adenoid grade, SBP z-score, and DBP z-score. Mean SpO2 was inversely associated with age, BMI z-score, NC, DBP z-score, and NLR, whereas minimal SpO2 was inversely associated with BMI z-score, NC, tonsil size, adenoid grade, SBP z-score, and DBP z-score.
In the preschoolers, the positive associations between AHI, tonsil size, and adenoid grade were significant. Otherwise, there were no statistically significant associations between AI, mean SpO2, minimal SpO2, and other variables of interest.
In the school-age children, AHI was positively correlated with BMI z-score, NC, tonsil size, SBP z-score, and DBP z-score. AI was positively associated with BMI z-score, NC, and DBP z-score. Mean SpO2 was inversely associated with BMI z-score, NC, and DBP z-score, whereas minimal SpO2 was inversely associated with BMI z-score, NC, SBP z-score, DBP z-score, and NLR.
In the adolescents, AHI was positively associated with BMI z-score, NC, and adenoid grade. Mean SpO2 was inversely associated with adenoid grade, whereas minimal SpO2 was inversely associated with BMI z-score, NC, and adenoid grade.’
Modified text, Pages 7-8, Table 2
Amended data.
Modified text, Page 9, Lines 212-220
Amended:
‘3.3 Variables Independently Associated with AHI, AI, Mean SpO2, or Minimal SpO2 in the Overall Cohort Using Multivariate Logistic Regression Analysis (Figure 1)
Figure 1 shows variables independently associated with disease severity parameters in the overall cohort. NC > 28.9 cm (p = 0.001), tonsil size > 3 (p = 0.008), and adenoid grade > 3 (p = 0.002) were independently correlated with AHI > 9.5 events/h (Figure 1a). NC > 28.9 cm (p = 0.001) and tonsil size > 3 (p = 0.044) were independently correlated with AI > 2.7 events/h (Figure 1b). NC > 28.9 cm (p < 0.001), tonsil size > 3 (p = 0.034), and DBP z-score > 0.59 (p = 0.009) were independently correlated with mean SpO2 < 97% (Figure 1c). BMI z-score > 0.86 (p < 0.001), tonsil size > 3 (p = 0.007), and DBP z-score > 0.59 (p = 0.023) were independently correlated with minimal SpO2 < 88% (Figure 1d).’
Modified text, Pages 9-10, Figure 1
Amended Figure 1 and corresponding legends.
Modified text, Page 10, Lines 234-253
Amended:
‘3.4 Independent Variables Associated with AHI, AI, Mean SpO2, or Minimal SpO2 in the Subgroups Using Multivariate Logistic Regression Analysis (Figure 2)
Figure 2 shows variables independently associated with disease severity parameters in the subgroups.
In the preschoolers, NC > 26.3 cm (p = 0.030) and tonsil size > 3 (p = 0.015) were independently correlated with AHI > 11.2 events/h (Figure 2a). Age > 5.0 years (p = 0.007) was independently correlated with AI > 3.6 vents/h (Figure 2b). SBP z-score > 0.25 (p = 0.003) was independently associated with mean SpO2 < 97% (Figure 2c). SBP z-score > 0.25 (p = 0.022) and DBP z-score > 0.81 (p = 0.045) were independently correlated with minimal SpO2 < 85% (Figure 2d).
In the school-age children, BMI z-score > 1.17 (p = 0.001) and tonsil size > 3 (p = 0.015) were independently correlated with AHI > 8.7 events/h (Figure 2e). BMI z-score > 1.17 (p = 0.014), tonsil size > 3 (p = 0.040), and DBP z-score > 0.38 (p = 0.017) were independently correlated with AI > 2.8 events/h (Figure 2f). NC > 29.8 cm (p = 0.009) was independently associated with mean SpO2 < 97% (Figure 2g). BMI z-score > 1.17 (p < 0.001) was independently correlated with minimal SpO2 < 88% (Figure 2h).
In the adolescents, adenoid grade > 3 (p = 0.007) was independently associated with AHI > 9.8 events/h (Figure 2i). Adenoid grade > 3 (p = 0.007) were independently correlated with AI > 2.2 events/h (Figure 2j). Tonsil size > 3 (p = 0.024) and adenoid grade > 3 (p = 0.024) were independently correlated with mean SpO2 < 96% (Figure 2k). Tonsil size > 3 (p = 0.037) and adenoid grade > 3 (p = 0.021) were independently associated with minimal SpO2 < 87% (Figure 2l).’
Modified text, Pages 11-12, Figure 2
Amended Figure 2 and corresponding legends.
Modified text, Page 12, Lines 276-283
Amended:
‘In this study, investigated how various factors differed among age subgroups, and how they were related to OSA severity. The three subgroups were similar in disease severity, which enabled us to observe inter-group differences in pathogenic factors. Correlations between independent variables and disease severity parameters demonstrated how they might have contributed as aggravating factors. This effort was important for understanding how the development and progression of pediatric OSA change with age. In the discussion, we will address our findings sequentially on domains of factors including gender, anthropometric measures, blood pressure, and inflammatory biomarkers.’
Modified text, Pages 12-13, Lines 300-310
Amended:
‘… no well-validated growth chart reference for pediatric NC as there is for pediatric BMI. On the other hand, the adenoid grade was higher in the preschoolers, suggesting that the enlargement of adenoid was a more important pathogenic factor in younger pediatric OSA patients. Interestingly, from the correlation analysis, we could see that disease aggravating factors were different among subgroups. The size of the tonsil and adenoid were the two dominant variables contributing to disease severity for preschoolers. For school-age students, weight status and neck circumference were more influential. For adolescents, the effects on disease severity were more universally distributed, suggesting that the exacerbation of OSA might be more multi-factorial in this population. Pathophysiological factors not only vary with age, but those factors involved in acquiring pediatric OSA may also be different from those involved in worsening the disease. Future studies are warranted to better understand the underlying mechanisms.’
Modified text, Page 13, Lines 330-341
Amended:
‘… Concordant with these phenotypic differences, we found that the risk factors of the three age subgroups were intrinsically distinct. As adenotonsillectomy and anti-inflammation therapies are both important treatments for pediatric OSA, every child patient must be treated with individually tailored plans. Not surprisingly, NC, tonsil size, and adenoid grade were associated with OSA severity in general, consistent with previous reports. However, our findings show that, while adenotonsillar hypertrophy is still the most distinguishable and well-documented risk factor for children with OSA, different variables seem to contribute to disease severity to different extents across age subgroups. For preschoolers, NC and tonsil size are most dominant for high AHI, and SBP and DBP are most dominant for low SpO2; the influence of BMI z-score on high AHI and AI, and low minimal SpO2 starts to emerge in school-age children; and for adolescents, adenoidal hypertrophy is the most dominant for high AHI and AI, and adenotonsillar hypertrophy is the most dominant for low mean and minimal SpO2 and should be monitored and managed.’
- Section 3.3 and 3.4 is extremely hard to read. It would be greatly advisable to present results as a table instead of the text and figure as it would be more transparent.
REPLY. Thank you for this very valuable suggestion. We’ve revised our Figure1, 2, and the corresponding descriptions on 3.3 & 3.4 to make it easier to read.
- The style of the manuscript should be improved: in many places’ repetitions occur, multiple/singular form is incorrectly used.
REPLY. Thank you for reminding us of the issue. We’ve went through the manuscript again carefully and correct a few mistakes. The manuscript was sent for English editing by a native speaker as well. The certificate is attached in our reply for your information.
- In the abstract, no aim of the study is described.
REPLY. We’ve revised our Abstract and added the study aim. Thank you very much for helping us to improve our manuscript.
Modified text, Page 1, Lines 32-35
‘Pediatric obstructive sleep apnea (OSA) is associated with adverse health outcomes, however little is known about the diversity of this population. This cross-sectional observational study retrospectively reviewed 253 Taiwanese children …’
à
‘Pediatric obstructive sleep apnea (OSA) is associated with adverse health outcomes; however, little is known about the diversity of this population. This retrospective study aims to investigate age-related differences in anthropometric and clinical features of this population. A total of 253 consecutive Taiwanese children …’
Reviewer 4 Report
The manuscript is about "Differences in Demographic, Anthropometric, Inflammatory, and Clinical Features among Pre-school Children, School-age Children, and Adolescents with Obstructive Sleep Apnea".
1. There are to many abbreviations in the text that make reading difficult.
Author Response
June 15, 2020
IJERPH
Dear Reviewer,
We appreciate reviewers’ valuable comments on the previous version of our manuscript. We have made a point-by-point response to the reviewers' comments and suggestions and marked up the changes made from the previous article file as a revised manuscript file. Please find the “response to reviewers” section attached at the end of the letter in which we address all the mentioned concerns. Careful revisions have been made in the manuscript according to the reviewers’ suggestions. Please see the attachment.
Thank you again for reviewing our paper. We look forward to your further information.
Sincerely yours,
|
Li-Ang Lee, M.D., M.Sc., F.I.C.S.
|
AUTHORS’ REPLY TO REVIEWERS’ COMMENTS:
Reviewer 4’s comments:
Comment#1
There are too many abbreviations in the text that make reading difficult.
Response#1:
Thank you very much for the comment. We’ve revised the manuscript extensively, especially in Results 3.3 & 3.4 to make it easier to read.
We are still using the following abbreviations though, since these terms are not uncommonly used in the research field. Also, as they all are mentioned in the manuscript for a considerable amount of times, writing them in full words will largely lengthen the manuscript
We appreciate your understanding. Please refer to the glossary below. We hope it can be informative and helpful. Thank you again.
AHI: apnea–hypopnea index; AI: apnea index; BMI: body mass index; DBP: diastolic blood pressure; NC: neck circumference; NLR: neutrophil-lymphocyte ratio; PLR: platelet-lymphocyte ratio; SBP: systolic blood pressure; SpO2: oxygen saturation measured by pulse oximetry.
Round 2
Reviewer 2 Report
The authors have replied to the answers. No more comments
Reviewer 3 Report
Instead of repeating "interquartile range" introduce an abbreviation IQR.
Please adjust Table 2 to split each column into 2 r and p separately.